PERSPECTIVE

# Reciprocal Inclusion of Microbiomes and Environmental Justice Contributes Solutions to Global Environmental Health Challenges

Mallory J. Choudoir,[a] Erin M. Eggleston[b]

[a]Department of Microbiology, University of Massachusetts Amherst, Amherst, Massachusetts, USA
[b]Department of Biology, Middlebury College, Middlebury, Vermont, USA

Mallory J. Choudoir and Erin M. Eggleston contributed equally to this work. Author order was determined based on academic seniority.

**ABSTRACT** Generations of colonialism, industrialization, intensive agriculture, and anthropogenic climate change have radically altered global ecosystems and by extension, their environmental microbiomes. The environmental consequences of global change disproportionately burden racialized communities, those with lower socioeconomic status, and other systematically underserved populations. Environmental justice seeks to balance the relationships between environmental burden, beneficial ecosystem functions, and local communities. Given their direct links to human and ecosystem health, microbes are embedded within social and environmental justice. Considering scientific and technological advances is becoming an important step in developing actionable solutions to global equity challenges. Here we identify areas where inclusion of microbial knowledge and research can support planetary health goals. We offer guidelines for strengthening a reciprocal integration of environmental justice into environmental microbiology research. Microbes form intimate relationships with the environment and society, thus microbiologists have numerous and unique opportunities to incorporate equity into their research, teaching, and community engagement.

**KEYWORDS** microbiome, environmental microbes, environmental justice, social equity, global change, stewardship, research ethics, microbiome stewardship

Microbes play monumental roles in global health. Our microbiomes are intimately linked at cellular scales to our physical and mental well-being and intrinsically linked to public health and social justice within a sociopolitical context (1, 2). Settler colonialism and neocolonialism substantially and permanently alter how humans interact with each other and with the environment (3–7). This includes land use change and dispossession, disruption of local food systems, erasure of Indigenous knowledge, forced migration and labor, and displacement of people, plants, and animals. We must not only consider how microbes magnify these anthropogenic impacts, but also how these same sociopolitical processes shape environmental microbiomes.

Environmental racism (see Historical Box), "refers to any policy, practice, or directive that differentially affects or disadvantages (whether intended or unintended) individuals, groups, or communities based on race or color" (8). Historically, the negative impacts of environmental policy disproportionately affect racialized and gendered communities and those of lower economic status (8–11). Power relationships within institutions and governments further impede equitable environmental policy (12). Environmental justice seeks to balance the relationship between environment and community, and to prioritize local communities in decisions concerning environmental laws, regulations, and policies. Definitions vary (13) but invoking environmental racism and justice has meaningful impacts and political implications. To address persistent

Address correspondence to Erin M. Eggleston, eeggleston@middlebury.edu.

The authors declare no conflict of interest.

> ### HISTORICAL BOX: ENVIRONMENTAL JUSTICE AND THE EPA
>
> The U.S. Environmental Protection Agency defines environmental justice as, "the fair treatment and meaningful involvement of all people regardless of race, color, national origin, or income with respect to the development, implementation, and enforcement of environmental laws, regulations, and policies (epa.gov/environmentaljustice)." This definition is noteworthy primarily due to the political capital it merits, but also in historical significance. It was not until 1991 that the EPA established a working group on environmental equity, nearly a decade after the Warren County, NC protest against the placement of a PCB landfill in a predominantly Black and underserved community. This protest is widely regarded as the genesis of the Environmental Justice movement in the United States (9).

disparities in underserved communities, solutions will vary since "people in different geographic, historical, political, and institutional contexts understand [environmental racism and environmental justice] differently (11)." Microbes are fundamentally embedded in the environment, and thus, are central to environmental justice issues. Simultaneously, microbes may also offer sustainable and effective solutions to environmental injustices.

**Environmental health and microbial injustice.** Environmental microbiomes (i.e., microbiomes predominately associated with soils and sediments, rivers and oceans, the built environment, and the atmosphere) sustain human life indirectly across planetary scales by supporting food systems, mediating biogeochemical cycles (14–16), and remediating waste (17). Importantly, microbes connect the health of human societies to the health of Earth's ecosystems, underpinning the concept of planetary health (18). Human activity has unequivocally altered the global climate system, ecological dynamics, and societal structures to such an extent as to usher in the "Anthropocene" as both an epoch of geology (19) and philosophy of thought (20, 21). Certainly, there are profound asymmetries in terms of individuals and communities that create anthropogenic environmental harm, and those that experience it (22, 23). Environmental microbiomes are themselves radically shaped by anthropogenic actions too (24–26).

Increasingly, we understand that disrupting environmental microbiomes amplifies social inequities, reinforces disproportionate access to natural resources, and perpetuates historical legacies of injustice. Across the urban to rural continuum, communities that rely on the local environment, due to cultural connection or necessity, are more susceptible to environmental harm (27–29). Green space (e.g., park, garden, arboretum) proximity and accessibility across global urban areas correlates with socioeconomic status, income, age, and education but inversely correlates with pollution exposure (30–32). Important health benefits associated with access to green spaces include improved mental and physical health and long-term reduction in mortality (32–34). Environmental microbiome exposures in green spaces relate to similar health benefits (35–37), and conversely, the biodiversity hypothesis suggests that reduction of environmental microbial exposures negatively impacts human health (38).

Modern microbiology research is fundamentally interdisciplinary, expansive, and innovative. Thus, microbiologists are uniquely positioned to build equity and justice into their work, and reciprocal engagement of environmental justice and microbiomes ensures impactful research. We urge microbiologists to build a more holistic knowledge of the relationships between the environment, microbiomes, and humanity.

## ENVIRONMENTAL MICROBES AND GLOBAL CHANGE

Global change significantly alters environmental microbiome composition, community resilience, and function (24, 39–41). Global change factors uncouple microbial biodiversity and ecological functions (42), which may negatively impact the resilience of

these ecosystems to increasingly frequent and catastrophic disturbance events. These disturbance events, like hurricanes or fires, are also sites of environmental injustice as they disproportionately burden underserved communities (27, 43). Long-term ecosystem and societal sustainability necessitates centering environmental microbes and understanding how they both contribute to and are affected by global change (26).

Soils harbor vast microbial biodiversity on which food, fiber, and fuel systems are built upon. Overuse, erosion, and land use change threaten soil biodiversity and their important ecosystem functions. For example, intensive agriculture is the leading cause of global land use change (44). Current food production practices exhaust natural resources and generate tremendous waste, and these systems are further jeopardized by ecosystem disturbances and climate change (45). Intensive agriculture practices homogenizes soil microbiomes across the globe (46), which can lead to potential genetic "spill over" into native nonagronomic ecosystems (47).

Soil microbes and sustainable land management practices may hold the key to global change mitigation strategies and food security. Centering soil biodiversity supports global sustainability agendas (48, 49). For instance, crop productivity and nutrition is driven by complex ecological and evolutionary dynamics of soil and plant microbiomes. Interdisciplinary research focusing on agroecosystem microbiomes may unlock a tool kit of solutions for sustainable agriculture (50–54).

Anthropogenic coastal processes (e.g., coastal mining and pollution) linked to food systems and industrialization are pervasive global injustices (55–57). Further, the marine carbon cycle is changing in response to ocean warming and acidification. To illustrate, microbial responses to these stressors result in shifts from cell growth to cell maintenance and other physiological stress adaptations, ultimately impacting global carbon fluxes due to decreased growth efficiency and altered carbon metabolism (26, 58).

Coral reef ecosystems are biodiversity hot spots linked to the carbon cycle through photosynthesis and their calcium carbonate structures (59). They are critical locally in place-based human 'reciprocal relationships' (60). Regionally, these ecosystems offer tourism and recreation opportunities, culture and identity, protection from storm events, and economic and food security linked to fisheries (61). Ocean warming and acidification impacts coral reef microbial communities both directly and indirectly (62, 63). Temperature change and other human disturbances disrupt the critical relationship between corals and their symbiotic photosynthetic microalgae and microbiomes (64, 65), leaving them susceptible to diseases (66). While human communities likely have short-term resilience and adaptability to these symptoms of climate change, more chronic long-term changes are predicted to exceed this resilience, impacting food security from local to global scales (67, 68).

## FRAMESHIFT TOWARD ENVIRONMENTAL MICROBIAL JUSTICE

Humanity is approaching a narrowing window of opportunity to adapt to anthropogenic global change impacts. Recently, a scientific cohort spanning diverse disciplines called for urgent, transformative, and equitable action to confront the climate crisis (69). Microbiologists are uniquely positioned to engage in collaborative thinking across fields to tackle knowledge gaps in environmental microbiology (70) and to address climate change (71). A paradigm shift advocates for direct integration of microbiology and social equity work (1, 2), as there are many axes along which microbiologists can broaden our research collaborations to more fully engage with social and environmental justice (72–75). We've compiled guidelines and resources (Table 1) to support progressive change within ourselves and within our communities to shift the field of microbiome science toward more environmentally just research. Below, we further contextualize ways in which we can prioritize equity and justice within environmental microbiome research.

**Guidelines for environmentally just microbiology.** Extractive research perpetuates colonial capitalist systems. "Helicopter research" is the harmful practice of scientists, usually from wealthier institutions, collecting samples from lower-income regions

**TABLE 1** Guidelines and resources for building more equitable environmental microbiome research practices targeted at individual, community, and institutional-level change

| Action Level | Action Step |
|---|---|
| Individual reflection | • Re-examine your own thinking around the connections between microbes and environmental, animal, human, and plant health. Recalibrate your relationship to microbes.<br>• Reframe our thinking around biological connections, from molecular to microbial to planetary scales (96).<br>• Critically reflect on who has the expertise and who is considered expert. Challenge the notions of what makes someone a scientist (82), or an environmental justice advocate, and uplift these individuals or communities. |
| Individual action | • Create and contribute to antiracist STEM communities (research groups, programs, departments, and beyond).<br>• Actively build anti-oppressive academic research groups (106). Set goals to build inclusivity and equity into microbiome research, collaborations, and training (107, 108).<br>• Contribute your expertise to social or environmental justice activism. How can you inform local, regional, national, or global scale policy change? With whom might you partner to advocate for and implement these changes?<br>• Exercise equitable citational practices. Who are you citing in your manuscripts (109, 110)? Gender balance (111)? Are you citing BIPOC scholars? How can you cite Indigenous knowledge (112)?<br>• Prioritize physical and mental safety when conducting field work for yourself, your colleagues, and any students or trainees with whom you work. Engage in inclusive, accessible, and safe field work (113, 114). |
| Community & research group | • Move beyond ethical research guidelines and towards dismantling colonial legacies in research institutions, projects, and within ourselves as scholars (115).<br>• Support, collaborate with, and hire BIPOC scholars. Invite BIPOC scholars to meetings and talks to share their research expertise. Representation matters in conference, departmental, keynote, panelist, and seminar speaker series.<br>• Practice ethical publication. Where are you publishing? Is your research open access? Are your data and code available and reproducible? How are you sharing your research with diverse audiences and stakeholders? Information access in environmental microbial ecology is still predominantly academic and privileged (116). Commit to basic, applied, and translational research that addresses power structures and broadens the scope and standards of our scholarship communication (117–119). This is especially important in communities historically excluded from this work. |
| Microbiome stewardship | • Consider how your personal, professional, and recreational activities alter microbial ecosystems.<br>• How do your individual and collective consumer behaviors relate to environmental microbiomes within built and managed environments? To the public health of your local communities?<br>• Engage in community-based research. Accept that there is a different timeline for this. Advocate for institutional and structural changes that prioritize engagement with and funding of this work.<br>• Build relationships with diverse stakeholders to more effectively generate questions, and to promote research that meaningfully engages with local and regional communities.<br>• Translating principles of community-based participatory research across disciplines offers guidance for research carried out with communities to effect change (120–123) in environmental microbiome research. |

or countries, processing, and analyzing data with little impact in local communities, and this dynamic is often experienced between the Global North and Global South (76, 77). Since microbes are small and numerous, environmental microbiome sampling trips may appear minimally invasive. However, field work has social, ecological, and intellectual implications beyond the collection site, including travel and lodging, ecological impacts of sampling, and extraction of local biogeochemical, molecular, or environmental data.

To shift our work toward a more community-minded ethos, microbial ecologists must critically examine the theoretical basis of our research questions, study designs, institutional legacies (78), and field sites. Decolonizing science seeks to deconstruct the political, social, and institutional colonialist foundations that continue to shape scientific practices (79, 80) while aspiring to expand ways of knowing and knowledge creation (81, 82). Anticolonial approaches for a more ethical field of ecology work to radically decolonize our minds, approaches to scientific practice, access to scientific resources, and expertise (83).

Microbial evolutionary and ecological intrinsic properties (e.g., beneficial plant-microbe interactions, microbial antibiotics and natural products, molecular biotechnology) are often harnessed and commodified without conversations surrounding data sovereignty. Microbial solutions to global health challenges should be developed and applied within an equitable framework. Productive conversations around human genomic ancestry, settler colonial concepts of people's origin, and Indigenous knowledge and reclamation has shifted the discussion around native DNA and kinship with

social, cultural, and political implications (84–88). These efforts provide a roadmap for necessary conversations about cultural, traditional, and scientific practices and data sovereignty regarding environmental samples and microbial culture collections on unceded Indigenous lands.

We acknowledge that these goals are impeded by systematic and structural barriers challenging to overcome. Beyond the inequity, racism, and sexism built into our academic institutions, current funding timelines discourage establishing strong meaningful relationships and compensating community members for their time and or expertise. However, by better understanding the history of microbiology and its legacy of colonialism, we can build anti-oppressive teams that support and uplift systematically underserved identities in knowledge building, experimental design and data collection, data sharing, and co-production. This type of research empowers not only the scientists, but the communities with whom we work (83).

**Environmental microbiome stewardship.** Environmental stewardship is the responsible use and protection of nature through conservation and sustainable practices that enhance ecosystem resilience and human well-being (89). As a parallel, the practice of antimicrobial (90, 91) and human gut microbiome (92) stewardship extends these principles to microbial life. We are not only intrinsically connected to our personal microbiomes, but also to the environmental microbiomes of our shared natural, engineered, and built habitats. Similarly, we should foster the stewardship of environmental microbiomes since preserving and maintaining our collective microbiomes is beneficial to all (92, 93).

Local and traditional knowledge applied in ecosystem and adaptive management strategies by Indigenous groups emphasizes the importance of stewardship for resilient ecosystems (94) and offers conservation strategies for curtailing macrobe biodiversity loss (95). These same strategies can extend to microbial conservation. Microbiome science is embedded within Indigenous knowledge systems, and environmental stewardship supports ecosystem resilience and establishes connections between humans and the planet (96). Indigenous sovereignty through land restitution (i.e., Land Back movement) not only has the potential to benefit human community health, but also minimizes cultural losses that parallel biodiversity loss (97–99).

Microbiome stewardship benefits human and non-human entities. For example, incorporating beneficial microbial exposure as a feature of urban landscape design supports community health (35–37). In agroecosystems, intensive agriculture and land management practices that contribute to soil erosion and nutrient loss are linked to detrimental human health outcomes (100). Soil degradation and anthropogenic climate change amplify global threats through positive feedback mechanisms (101). Within the past few decades in the southwestern United States, a massive increase in "Valley fever" outbreaks correlates to more frequent and intense dust storms (102, 103). Valley fever is a potentially life-threatening pulmonary infection caused by the inhalation of airborne spores of the fungus *Coccidiodes immitis* (104). A 2018 study reported elevated occupational health risks and Valley fever cases in women and Hispanic agricultural workers (105). Stewardship and agronomic practices that minimize soil erosion (e.g., crop rotations, cover crops, low tillage, and land contouring) also shape soil and atmospheric microbiomes with cascading human health impacts. This illustrates how inclusion of microbiome science relates to both global issues, such as food security, and also local social equity conditions and agricultural worker's rights.

## CONCLUSION

As we advance our understanding of the evolutionary and ecological dynamics of environmental microbiomes in a world continuously shaped by global change, we must foreground environmental justice. It is an important and exciting time to be a microbiologist. Our work must ensure equity from the development and design of projects to effective and inclusive communication, and finally to implementation of applications and policies derived from our research. By centering environmental

microbiomes at the root of global change and environmental justice, we can shift our collective experiences to one based upon microbial justice.

## ACKNOWLEDGMENTS

We appreciate the Microbes and Social Equity (MSE) working group (2) for community and collaboration. We thank Drs. Lesley-Ann Giddings, Hemangini Gupta, Sheila Saia, Daniel Suarez, and Kristen DeAngelis, as well as three anonymous reviewers, for their constructive feedback on this manuscript. M.J.C. and E.M.E. are a team of two white women, academic microbiologists in the United States who strive to further align our research with environmental justice. Our identities are important to note, as we are privileged beneficiaries of historical practices that continue to support white scientists. We acknowledge that our work is conducted on unceded Indigenous lands. We honor the long history of stewardship that shapes these ecosystems, and we commit to relationship building and research that works toward a sustainable and equitable future. M.J.C. is supported by the National Science Foundation Division of Environmental Biology under grant 1749206.

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
