## [Reviewer comments · mSystems]

Reciprocal inclusion of microbiomes and environmental justice can contribute solutions to global environmental health challenges

Mallory Choudoir and Erin Eggleston

Corresponding Author(s): Erin Eggleston, Middlebury College

Review Timeline:

Submission Date:	January 7, 2022
Editorial Decision:	February 1, 2022
Revision Received:	April 1, 2022
Accepted:	May 5, 2022

Editor: Suzanne Ishaq

Reviewer(s): Disclosure of reviewer identity is with reference to reviewer comments included in decision letter(s). The following individuals involved in review of your submission have agreed to reveal their identity: Kelly Ramirez (Reviewer #1); María Rebolleda Gómez (Reviewer #2); Samiksha Raut (Reviewer #3)

Transaction Report:

DOI: <https://doi.org/10.1128/msystems.01462-21>

February 1, 2022

Dr. Erin M Eggleston
Middlebury College
Department of Biology
McCardell Bicentennial Hall
Middlebury, VT 05753

Re: mSystems01462-21 (An invitation to consider the microbial perspective of environmental justice)

Dear Dr. Erin M Eggleston:

Thank you for submitting your manuscript to mSystems. We have completed our review and I am pleased to inform you that, in principle, we expect to accept it for publication in mSystems. However, acceptance will not be final until you have adequately addressed the reviewer comments.

The reviewers and I am unanimous in our interest for this topic and this manuscript, and think that this highlights an important perspective in research. We are also unanimous in our recommendation for more specificity and detail in the piece, especially with regards to providing examples for the recommendations. Often with this type of perspective, the readers will agree with the message but not know where to begin to implement changes in their own work, thus more direction from the authors who are experienced in this interdisciplinary work would improve the impact of the paper.

Preparing Revision Guidelines

Sincerely,

Suzanne Ishaq

Editor, mSystems

Journals Department
Reviewer comments:

Reviewer #1 (Comments for the Author):

I really enjoyed the topic of this paper. I think more background would really increase accessibility to those unfamiliar with social justice work - rather than jumping straight into the microbes. Overall my comments are about organization and including more specifics/background. Thank you for tackling this important topic!

Title: 'Invitation' seems a bit passive, what about something more like - Inclusion of the microbiome can contribute solutions for environmental justice and global challenges

Abstract

L11: deeply? I think in the first sentence change intrinsic to critical and in this sentence "Because of these relationships to society, microbes are intrinsically embedded within..."

L17: Remove 'we invite' and make more broad, "Consideration of scientific and technological advances in environmental justice has become an important component of developing equitable and actionable solution to global equity challenges. Environmental microbiomes are intrinsically linked to environmental and human health. Through these relationships, microbes are deeply embedded within social and environmental justice. Here we identify a number of areas where inclusion of microbial research and knowledge can further benefit global equity goals. We also offer guidelines for more broadly...."

L19: Perhaps move the first 2 sentences before finally, so all 'microbiome' content is together.

L20: Microbes form intimate relationships with the environment and society, thus microbiologists have numerous opportunities to incorporate values of equity into their research and dissemination of those results.

Intro - overall this section needs to better center on the value of including social justice frameworks into science/biodiversity. For someone unfamiliar with social justice this section doesn't go enough into the background and gets too quickly into microbiome. Save that for later.

L29: Delete 'despite small size' this would be more for general article rather than one for msystems.

L44: specify what 'unequal access' means - and what those benefits are.

L45-52: I would like to see this background info to be more specific, and also maybe give examples of how other forms of biodiversity have been included in a social justice framework.

L55: Solutions for what?

L55-56: Give more context for why this is important for the reader to know

L53-62: this paragraph could be rewritten to have better flow.

L60-62: But be more specific about why this is so critical and why this is a good time and why microbiologists/scientists should take up this role.

2

L71-73: unclear what point is being made, perhaps it is just out of place and would be better elsewhere?

3 - this section is great. Might be worth adding a section on human diseases and health as a response to global change?

4

L151: perhaps be explicit about who/when can implement recommendations

L165: beyond how?

Reviewer #2 (Comments for the Author):

The authors in this paper argue that social injustices have led to differences in environmental microbiomes in forms that can affect health. As such, they propose ways to move forward by making changes in microbial ecology. I strongly sympathize with the authors' intentions and I think their message is important and timely. However, I think the paper as it stands is hard to differentiate from other papers suggesting decolonization and more inclusion.

I agree with the authors that there is particular importance in including the microbiome in environmental justice conversations, but I think this argument needs to be further developed: expand on how environmental impacts can affect the microbiome and thus our health. I think there are few papers showing the whole chain (and that is important to remark). There are papers showing that environmental impacts are not evenly distributed, papers showing that anthropogenic impact on the environment

affects the microbiome, and others showing that microbiome changes affect health. What is the impact industrial farmers have on the microbiome and health of workers? Can the microbiome help workers demand better working conditions? Industrial agriculture, for example, has clearly disturbed soil microbiomes, increased erosion, and drought. The combination of all of those factors has increased the incidence of respiratory diseases in the San Joaquin Valley in California, in particular, "valley fever". How can microbiome research help predict and prevent the emergence of new diseases in marginalized communities, affected by extractive economies?

In particular, I found the recommendations too general (and therefore vague). In terms of colonization for example, how can we facilitate the stewardship of practices that might sustain different microbiomes? Can we provide resources for indigenous research in microbiomes, similar to the efforts to empower indigenous populations to study and steward their own genetic information?

Minor comments

L. 36 - It sounds weird to say that human activity has changed ecology and sociology. It has changed ecological dynamics and the structure of societies.

L. 38 - Some people refer to the Capitalocene to make clear that it is not all humans to the same degree (see for example Raj Patel's -7 cheap things).

Reviewer #3 (Comments for the Author):

The article entitled "An invitation to consider the microbial perspective of environmental justice" is well written and highlights the need to take into account environmental justice perspective. The article begins with a robust introduction to the topic by presenting the historical framework enabling a reader to dive further to finally understand the proposed recommendations.

Based upon my review, I would like to suggest a few minor changes :

(1) Positionality of the authors needs to be separated into a separate section beneath the introduction instead of being a part of the introduction. A few sentences related to self reflection and/or affiliation with a social equity workgroup, etc. as applicable should help the reader understand why the authors decided to embark on writing this article.

(2) Abstract makes a mention of guidelines whereas the article makes a mention of recommendations. However, after reading the recommendations section, it gave me an impression of guidelines rather than concrete recommendations. Addition of concrete actionable steps would leverage this article. Maybe addition of a table to indicate a summary ?

(3) Section C "Ethical and just research depends on building human relationships" - I would recommend adding how involvement of minoritized faculty/individuals who help drive the change.

Dear Dr. Ishaq and reviewers,

We sincerely thank the reviewers for their constructive comments, which have greatly improved this manuscript. Overall, we agree with reviewer feedback and have made substantial changes to the order in which we present our major themes and the specificity of our examples. For instance, in the Introduction we moved the section framing social and environmental justice to before the section on environmental microbiomes. This also necessitated significant changes to our transitions and order throughout the manuscript, and we have updated those for clarity.

Most dramatically, we revamped the final section to be more explicit and actionable. We added a table referencing specific guidelines and resources, which addresses individual, community, and institutional-level actions. This section was revised to further illustrate and contextualize the recommendations in the table.

In addition to restructuring, we have addressed individual reviewer comments for small to medium concerns as well. We acknowledge that this manuscript resubmission is significantly revised, but that the main objectives and themes from the original manuscript are presented more clearly and with a stronger voice. (Please note this Response to Reviewers utilizes the line numbering of the new manuscript).

Sincerely,
Erin and Mallory

Reviewer comments:

Reviewer #1

I really enjoyed the topic of this paper. I think more background would really increase accessibility to those unfamiliar with social justice work - rather than jumping straight into the microbes. Overall my comments are about organization and including more specifics/background. Thank you for tackling this important topic!

We now begin our introduction section with an overview of social and environmental justice concepts. We agree that priming this information for the reader is likely to assist with accessibility to a microbially-focused readership.

Title: 'Invitation' seems a bit passive, what about something more like - Inclusion of the microbiome can contribute solutions for environmental justice and global challenges

Our new title is less passive, "Reciprocal inclusion of microbiomes and environmental justice can contribute solutions to global environmental health challenges."

Abstract

L11: deeply? I think in the first sentence change intrinsic to critical and in this sentence "Because of these relationships to society, microbes are intrinsically embedded within..."

We have rearranged the abstract in keeping with the changes to the body of the text. This sentence no longer exists.

L17: Remove 'we invite' and make more broad, "Consideration of scientific and technological advances in environmental justice has become an important component of developing equitable and actionable solution to global equity challenges. Environmental microbiomes are intrinsically linked to environmental and human health. Through these relationships, microbes are deeply embedded within social and environmental justice. Here we identify a number of areas where inclusion of microbial research and knowledge can further benefit global equity goals. We also offer guidelines for more broadly....

We agree, this was a bit narrowly focused initially. We have updated the abstract conclusion with your L20 feedback to reflect a broader welcoming call to action.

L19: Perhaps move the first 2 sentences before finally, so all 'microbiome' content is together.

Yes, we agree this order better aligns the microbiome content. We have changed some of the specific language here.

L20: Microbes form intimate relationships with the environment and society, thus microbiologists have numerous opportunities to incorporate values of equity into their research and dissemination of those results.

This is a fantastic sentence and we have now included a slightly modified version. Thank you, and we hope that our paraphrasing is acceptable.

Intro - overall this section needs to better center on the value of including social justice frameworks into science/biodiversity. For someone unfamiliar with social justice this section doesn't go enough into the background and gets too quickly into microbiome. Save that for later.

Agreed. We revised this section which now starts with the social and environmental justice framing.

L29: Delete 'despite small size' this would be more for general article rather than one for msystems.

We agreed with the recommended word choice, and removed this phrase.

L44: specify what 'unequal access' means - and what those benefits are.

We cut "unequal access" and expanded the urban green space discussion, see next comment.

L45-52: I would like to see this background info to be more specific, and also maybe give examples of how other forms of biodiversity have been included in a social justice framework. *This green space example has been expanded to provide more specifics, and additional social justice/biodiversity examples are now included in the Environmental Microbiome Stewardship section. L 70-74*

L55: Solutions for what?

This sentence was cut.

L55-56: Give more context for why this is important for the reader to know

As per Reviewer #3, we have moved our positionality statement to Acknowledgements, and updated the need for its inclusion in the piece.

L53-62: this paragraph could be rewritten to have better flow.

Agreed, it was reworked in alignment with the revised introduction section.

L60-62: But be more specific about why this is so critical and why this is a good time and why microbiologists/scientists should take up this role.

We have revised the section "Frameshift towards environmental justice" L124, to touch on this critical time for microbiology, climate change, and social equity.

Section 2

L71-73: unclear what point is being made, perhaps it is just out of place and would be better elsewhere?

This sentence was cut, and the clarified idea was added into the environmental justice introductory section. L36-38 "We must not only consider how microbes magnify anthropogenic impacts, but also how these same sociopolitical processes shape environmental microbiomes."

Section 3 - this section is great. Might be worth adding a section on human diseases and health as a response to global change?

Yes, we wholeheartedly agree with the importance of global change and human disease! However, we intentionally chose to center our discussion on non-human associated environmental microbiomes, as this is our area of research expertise.

Section 4

L151: perhaps be explicit about who/when can implement recommendations

The recommendations and guidelines are now split between textual examples and a table with different levels of entry for implementation.

L165: beyond how?

Clarified, L144-146 "However, field work has social and environmental implications beyond the field plot and collection tube, including travel and lodging to and from the field site, ecological impacts of sampling, and extraction of local biological data."

Reviewer #2

The authors in this paper argue that social injustices have led to differences in environmental microbiomes in forms that can affect health. As such, they propose ways to move forward by making changes in microbial ecology. I strongly sympathize with the authors' intentions and I think their message is important and timely. However, I think the paper as it stands is hard to differentiate from other papers suggesting decolonization and more inclusion.

I agree with the authors that there is particular importance in including the microbiome in environmental justice conversations, but I think this argument needs to be further developed: expand on how environmental impacts can affect the microbiome and thus our health. I think there are few papers showing the whole chain (and that is important to remark). There are papers showing that environmental impacts are not evenly distributed, papers showing that anthropogenic impact on the environment affects the microbiome, and others showing that microbiome changes affect health. What is the impact industrial farmers have on the microbiome and health of workers? Can the microbiome help workers demand better working conditions? Industrial agriculture, for example, has clearly disturbed soil microbiomes, increased erosion, and drought. The combination of all of those factors has increased the incidence of respiratory diseases in the San Joaquin Valley in California, in particular, "valley fever". How can microbiome research help predict and prevent the emergence of new diseases in marginalized communities, affected by extractive economies?

In particular, I found the recommendations too general (and therefore vague). In terms of colonization for example, how can we facilitate the stewardship of practices that might sustain different microbiomes? Can we provide resources for indigenous research in microbiomes, similar to the efforts to empower indigenous populations to study and steward their own genetic information?

Thank you for highlighting this weakness of the manuscript, and for these specific thematic suggestions connecting environmental microbiome research to workers rights and Indigenous knowledge sovereignty. We have updated the manuscript to include these topics, Valley Fever in L197-201 and ownership of native environmental microbiome diversity/knowledge in L156.

Minor comments

L. 36 - It sounds weird to say that human activity has changed ecology and sociology. It has changed ecological dynamics and the structure of societies.

We have revised this sentence as, "Human activity has unequivocally altered the global climate system, ecological dynamics, and societal structures..."

L. 38 - Some people refer to the Capitalocene to make clear that it is not all humans to the same degree (see for example Raj Patel's -7 cheap things).

Yes, this is an important point. We have highlighted this on L62-64 "Certainly, there are profound asymmetries in terms of individuals and communities that create anthropogenic environmental harm, and those that experience it (22, 23)."

Reviewer #3 (Comments for the Author):

The article entitled "An invitation to consider the microbial perspective of environmental justice" is well written and highlights the need to take into account environmental justice perspective. The article begins with a robust introduction to the topic by presenting the historical framework enabling a reader to dive further to finally understand the proposed recommendations. Based upon my review, I would like to suggest a few minor changes :

(1) Positionality of the authors needs to be separated into a separate section beneath the introduction instead of being a part of the introduction. A few sentences related to self reflection and/or affiliation with a social equity workgroup, etc. as applicable should help the reader understand why the authors decided to embark on writing this article.

We updated our reflection on positionality and moved it to the Acknowledgement section.

(2) Abstract makes a mention of guidelines whereas the article makes a mention of recommendations. However, after reading the recommendations section, it gave me an impression of guidelines rather than concrete recommendations. Addition of concrete actionable steps would leverage this article. Maybe addition of a table to indicate a summary ?

Thank you for this suggestion. We have separated out a table of concrete actionable steps as guidelines. The text for the section, "Frameshift towards environmental microbial justice," now highlights two contextual vignettes through which microbiologists might engage with this work.

(3) Section C "Ethical and just research depends on building human relationships" - I would recommend adding how involvement of minoritized faculty/individuals who help drive the change.

We have included this recommendation in Table 1

May 5, 2022

Dr. Erin M Eggleston
Middlebury College
Department of Biology
McCardell Bicentennial Hall
Middlebury, VT 05753

Re: mSystems01462-21R1 (Reciprocal inclusion of microbiomes and environmental justice can contribute solutions to global environmental health challenges)

Dear Dr. Erin M Eggleston:

The reviewers and I agree that the revisions to the manuscript have strengthened this piece, and I am pleased to accept it for publication. Reviewer one suggested some minor rephrasing which may be helpful, and which could be done during the article proofing stage as needed.

Your manuscript has been accepted, and I am forwarding it to the ASM Journals Department for publication. For your reference, ASM Journals' address is given below. Before it can be scheduled for publication, your manuscript will be checked by the mSystems production staff to make sure that all elements meet the technical requirements for publication. They will contact you if anything needs to be revised before copyediting and production can begin. Otherwise, you will be notified when your proofs are ready to be viewed.

Publication Fees:

We recognize that the video files can become quite large, and so to avoid quality loss ASM suggests sending the video file via <https://www.wetransfer.com/>. When you have a final version of the video and the still ready to share, please send it to mSystems staff at mssystems@asmusa.org.

For mSystems research articles, if you would like to submit an image for consideration as the Featured Image for an issue, please contact mSystems staff at mssystems@asmusa.org.

Sincerely,

Suzanne Ishaq
Editor, mSystems

Journals Department
Phone: (202) 737-3600